# Satellite Image Multi-Frame Super Resolution Using 3D Wide-Activation Neural Networks

**Francisco Dorr**

Independent Researcher, José María Paz 745 12, Florida 1602, Argentina; fdorr@dc.uba.ar

**Abstract:** The small satellite market continues to grow year after year. A compound annual growth rate of 17% is estimated during the period between 2020 and 2025. Low-cost satellites can send a vast amount of images to be post-processed at the ground to improve the quality and extract detailed information. In this domain lies the resolution enhancement task, where a low-resolution image is converted to a higher resolution automatically. Deep learning approaches to Super Resolution (SR) reached the state-of-the-art in multiple benchmarks; however, most of them were studied in a single-frame fashion. With satellite imagery, multi-frame images can be obtained at different conditions giving the possibility to add more information per image and improve the final analysis. In this context, we developed and applied to the PROBA-V dataset of multi-frame satellite images a model that recently topped the European Space Agency's Multi-frame Super Resolution (MFSR) competition. The model is based on proven methods that worked on 2D images tweaked to work on 3D: the Wide Activation Super Resolution (WDSR) family. We show that with a simple 3D CNN residual architecture with WDSR blocks and a frame permutation technique as the data augmentation, better scores can be achieved than with more complex models. Moreover, the model requires few hardware resources, both for training and evaluation, so it can be applied directly on a personal laptop.

**Keywords:** multi-frame super resolution; wide activation super resolution; 3D convolutional neural network; deep learning

## 1. Introduction

In the past, the satellite market was reserved for a few companies and governments, which had the capacity (technical and monetary) to build and deploy large machinery in space, and the data obtained afterwards were used just by only a few research teams worldwide. Today, there is a growing interest, both social and commercial, in the deployment of small, low-cost satellites. A compound annual growth rate of 17% has been estimated for the small satellite market (forecast from 2020 to 2025) [1]. This expansion brings with it new challenges because of the vast amount of new data available. For example, satellite images are used in many different fields to accomplish a wide spectrum of tasks. To name a few, Xu et al. [2] investigated vegetation growth trends over time; Martinez et al. [3] tracked tree growth through soil moisture monitoring; Ricker et al. [4] studied Arctic ice growth decay; and Liu et al. [5] developed a technique to extract deep features from high-resolution images for scene classification.

However, as satellites get more affordable and smaller, data quality cannot always be maintained; a trade-off must be found between price and quality. A case of study is High-Resolution (HR) images. They are not easy to obtain, or fast enough to transfer, and need costly and massive platforms as opposed to small, rapidly deployed, low-cost satellites that can provide viable services at the cost of lowering quality [6]. Image quality restrictions are common due to degradation and compression in the imaging process [7]. Downlink bandwidth is a primary concern for Earth Observation (EO) satellites,

and the improvement of data-rate transmission is an open research field. In 2012, the Consultative Committee for Space Data Systems (CCSDS) proposed a standard (131.2-B-1) [8] for high rate telemetry applications. Multiple works can be found addressing this problem; to name a few, Wertz et al. [9] developed a flexible high data rate transmitter operating in the X-band and Ka-band to maximize data throughput; Betolucci et al. [10] performed a trade-off analysis for 131.2-B-1 and its extension ModCod(SCCC-X), showing a 35.5% increase in maximum throughput. This notwithstanding, many tasks can be solved in post-processing steps, improving the quality once the data arrive on Earth, hence the importance of image resolution enhancement techniques that can take advantage of the huge and growing amount of information available from small satellites.

The problem of Super Resolution (SR) is not new. It has been widely studied in different contexts taking multiple approaches. Specifically, to improve the resolution of satellite images, one common approach is the use of Discrete Wavelet Transforms (DWTs), in which the input image is decomposed into different sub-bands and then combined to generate a new resolution through the use of inverse DWTs [11–13]. In recent years, as deep learning applications explode in every computer-vision task, convolutional neural network methods began to dominate the problem of SR. However, most of them focus on Single-Image Super Resolution (SISR) [14–16] and do not take advantage of the temporal information inherent to multi-frame tasks. MFSR has been studied in video; for example, Sajjadi et al. [17] proposed a framework that uses the previously inferred HR estimate to super resolve the subsequent frame; Jo et al. [18] created an end-to-end deep neural network that generates dynamic upsampling filters and a residual image avoiding explicit motion compensation; and Kim et al. [19] presented 3DSRnet, a framework that maintains the temporal depth of spatio-temporal feature maps to capture nonlinear characteristics between low- and high-resolution frames. Regarding the deep learning approaches to MFSR in satellite imagery, recent work has demonstrated its applicability: Märtens et al. [20] proposed a CNN, capable of coping with changes in illumination and cloud artifacts, that was applied to multi-frame images taken over successive satellite passages over the same region. Molini et al. [21] implemented a method that exploits both spatial and temporal correlations to combine multiple images, and Deudon et al. [22] created an end-to-end deep neural network that encodes, fuses multiple frames, and finally, decodes an SR image.

In this technical note, we present a technique that takes as a strong baseline the 3DSRnet framework of Kim et al. [19], but adapted for satellite imagery MFSR and replacing 3DCNN blocks with wide activation blocks [23]. This method's core is a 3D wide activation residual network that was fully trained and tested on the PROBA-V dataset [20] on a low-specification home laptop computer with only 4GB of GPU memory. Despite being low on resources and based on a simple architecture, this method topped Kelvin's ESA challenge in February 2020 [24].

## 2. Materials and Methods

### 2.1. Image Dataset

In this work, we used the set of images from the vegetation observation satellite PROBA-V of the European Space Agency (ESA) [20] provided in the context of the ESA's super resolution competition PROBA-V, which took place between 1 November 2018 and 31 May 2019 [25].

The PROBA-V sensors can cover 90% of the globe every day with a resolution of 300 m (low resolution). Every 5 days, they can provide images of 100 m resolution (high resolution). With this in mind, the objective of the challenge is to build the 100 m resolution images from multiple images of a higher frequency of 300 m resolution. It should be noted that the images provided for this challenge were not artificially degraded. As a common practice in super resolution developments, usually, a high resolution image is artificially degraded, and this is used as the low-resolution starting point. In this case, all the images are original, both the low- and high-resolution ones [25].

### 2.1.1. Dataset Characteristics

As described in Märtens et al. [20], the dataset used for both training and testing is composed as follows:

- One-thousand one-hundred sixty images from 74 hand-selected regions were collected at different points in time.
- This was divided into two spectral bands: RED with 594 images and NIR with 566; a radiometrically and geometrically corrected top-of-atmosphere reflectance in plate carrée projection was used for both bands.
- The LR size is $128 \times 128$ and the HR size $384 \times 384$, and both have a bit-depth of 14 bits, but saved as 16-bit png format.
- Each scene has a range of LR images (from a minimum of 9 to a maximum of 30) and one HR image.
- The mean geolocation accuracy of PROBA-V is about 61 m, and (sub-)pixel shifts can occur as the images are delivered as recorded by PROBA-V (not aligned with each other). This induces a possible 1 pixel shift between the LR frames of an image.
- For each LR and HR image, there is a mask that indicates which pixels can be reliably used for reconstruction. The masks were provided by Märtens et al. [20] and are based on a status map containing information on the presence of artifacts (clouds, cloud shadows, and ice/snow) generated from ESA's Land Cover Climate Change Initiative and ESA's GlobAlbedo surface reflectance data. The exact procedure on how this map is generated can be found in Section 2.2.3 of the PROBA-V product manual [26].

### 2.2. Network Architecture

The proposed 3DWDSRnet method for super resolution is based on a patch based 3D-CNN architecture that allows multiple image inputs to be scaled into a single higher resolution image.

The problem being investigated is very similar to video SR, where the resolution of a single video frame is enhanced using the information from the surrounding frames. In a given sliding time window, video frames usually refer to a single scene, but with subtle changes between each other. Thus, this temporal information can benefit resolution scaling more than single-image approaches [19]. The PROBA-V dataset has multiple frames per location, which can have shifts of up to one pixel. This evokes a similarity with the frames of a video and their possible variations, and that is why we decided to investigate this path.

Our work takes as a strong baseline the framework proposed by Kim et al. [19] for video super resolution: 3DSRnet. They used a 3D-CNN that takes five low-resolution input frames and seeks to increase the resolution of the middle frame. The network is a two way residual network. The main path acts as a feature extractor from the chaining of multiple 3D convolutional layers that preserves the temporal depth. For the last layers, a depth reduction is performed to obtain the final 2D HR residual. Meanwhile, the second path takes the middle frame and applies a bicubic scaling. A pixel shuffle [27] reshapes the residual output of the main path, which is then added to the output of the secondary path obtained by this means, the final HR frame (Figure 1).

Our approach differs from the original 3DSRnet in two main aspects (compare Figures 1 and 2):

- Convolutional layers are replaced by 3D WDSR blocks
- Bicubic upsampling is replaced by 2D WDSR blocks.

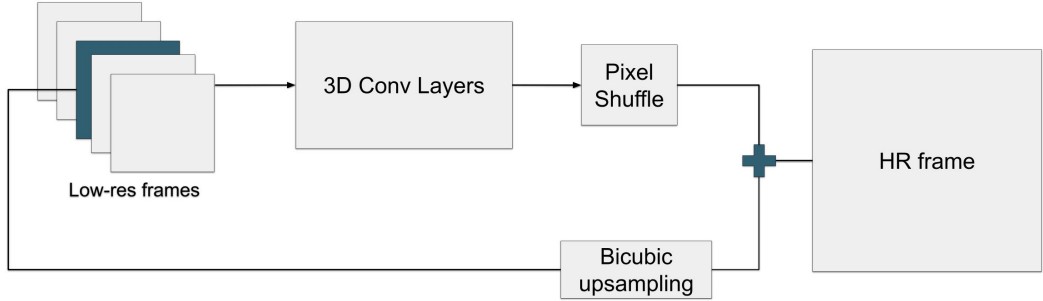

**Figure 1.** Original 3DSRnet: All low-resolution input frames are fed into the main 3D Conv path to predict the residuals for the middle frame. The results of the paths are added up to obtain the final HR frame.

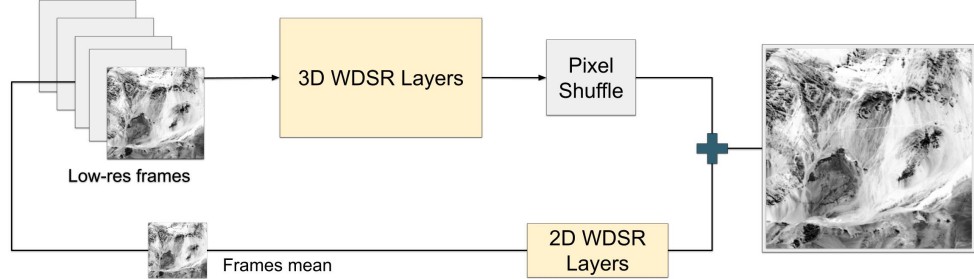

**Figure 2.** 3DWDSRnet: All low-resolution input frames are fed into the main WDSR Conv 3D path to predict the scene's residuals. The average of the frames is used as the input to the WDSR 2D Convs path. The results of both paths are added together to obtain the final HR frame. Soft yellow highlights the differences with the original 3DSRnet blocks.

### 2.2.1. WDSR Blocks

Yu et al. [23] described WDSR blocks as residual blocks with the capability to increase the final accuracy of an SISR task. They demonstrated that a feature expansion using a $1 \times 1$ Conv before the ReLU activation, followed by a feature factorization given by a $1 \times 1$ Conv and a $3 \times 3$ Conv, keeps more information and even lowers the number of total parameters used (Figure 3). This residual block was named WDSR-b. In our study, we expand the notion of the WDSR-b block from 2D to 3D and replace every single 3D Conv from 3DSRnet with it. To do so, we simply change the kernel sizes from $1 \times 1$ to $1 \times 1 \times 1$ and from $3 \times 3$ to $3 \times 3 \times 3$. Everything else remains exactly the same. The implementation of the WDSR block was based on Krasser's GitHub code [28].

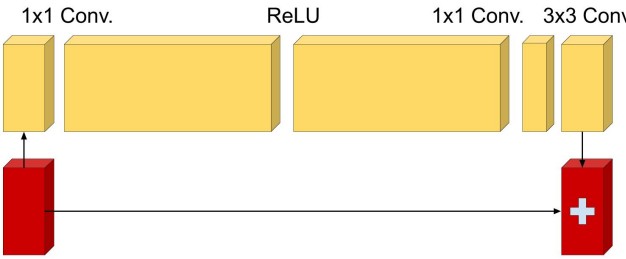

**Figure 3.** 2D-WDSR-bblock. The residual block is composed of a $1 \times 1$ Conv to expand features before ReLU activation. After activation, a $1 \times 1$ Conv followed by a $3 \times 3$ Conv are applied. 3D-WDSR-b used by our method follows the same approach, but taking into account the time dimension.

### 2.3. Preprocessing and Data Augmentation

The preprocessing steps were performed as follows:

1. Register all frames from each image to the corresponding first frame by phase cross-correlation to align the possible pixel-shifts (see the dataset characteristics in Section 2.2.1). We used the implementation from scikit-image: phase_cross_correlation [29].
2. Remove images where all of their frames had more than 15% dirty pixels. A dirty pixel is a pixel that has a mask showing the presence of an artifact, as explained in Section 2.2.1.
3. Select $k$ best frames ($k = 7$). To do this, we sort all the frames from the cleanest to the dirtiest (as a percentage of the total number of pixels) based on the masks provided and select the first $k$ frames.
4. Extract 16 patches per image.
5. Remove instances where the HR target patch had more than 15% dirty pixels.

To make the training more robust to the pixel shifts and differences between frames, a frame based data augmentation was performed. For each patch, six new patches were added to the training set, each of them with a random frame permutations. A similar augmentation technique can be found in Deudon et al. [22]. The impact of patches and data augmentation by frame permutation can be seen in Section 3.

### 2.4. Training

The training was performed on a low-end laptop GPU GTX1050 with 4GB of memory. First, the model was trained on the NIR band until no more improvements were found (156 epochs). Then, the RED band was trained over the NIR band pretrained model (61 epochs). This two-step model training was based on Molini et al. [21], where they found it increased the final accuracy.

We used the NAdam optimizer with a learning rate of $5 \times 10^{-4}$ a patch size of 34 with a stride of 32, and a batch size of 32 patches. The main path was composed of 8 3DWDSR-b blocks before the time dimension reduction. As regularization, it is important to note that common techniques such as batch normalization do not work well in SR problems ([30,31]). We used weight normalization as recommended by Yu et al. [23].

Quality Metric and Loss Function

Märtes et al. [20] proposed the quality metric clean Peak Signal-to-Noise Ratio (cPSNR) that takes into account the pixel shifts between frames and is applicable for images that only have partial information (dirty pixels). Basically, cPSNR ignores masked pixels (due to the wrong pixels, clouds, etc.) and takes into account all possible pixel shifts between predicted SR and HR to calculate the final metric. Inspired by cPSNR, Molini et al. [21] proposed cMSE, a modified Mean Squared Error to use as loss functions. Since SR prediction and the HR target could be shifted, the loss embeds a shift correction. Taking into account what was stated in Section 2.1.1 about PROBA-V mean geolocation accuracy (61 m), the maximum pixel shift $d$ between SR and HR is 3. Then, SR is cropped at the center by $d$ pixels, and all possible patch shifts $HR_{u,v}$ are extracted from the target HR image. Thereafter, all possible MSE scores are calculated for each $HR_{u,v}$ patch, and the minimum score is taken.

We found that this loss works quite well for the problem, but based on Zhao et al. [32], we follow their recommendation to use the Mean Absolute Error loss (MAE) as a substitute. Mathematically, cMAE is defined as follows:

$$L = \min_{u,v \in [0,2d]} \frac{\sum_{i=1}^{N_{u,v}} |HR_{u,v} - (SR_{crop} + b)|}{N_{u,v}}$$

where $N_{u,v}$ is the total number of clean pixels in the $u, v$ crop and $b$ is the brightness bias corrections. Figure 4 shows how the loss is calculated taking into account all possible pixel shifts.

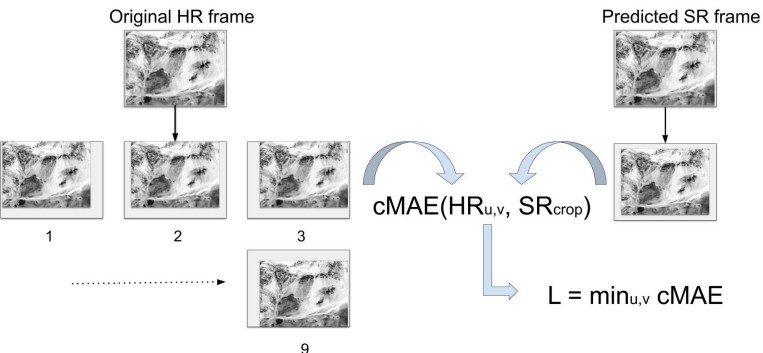

**Figure 4.** cMAE. For each possible pixel shift u,v, the cMAE is calculated between the cropped SR and the $HR_{u,v}$ patch. In this example figure, the maximum possible shift is 1 (both horizontally and vertically), so 9 patches are extracted for each possible combination. The PROBA-V dataset has a maximum of 3 pixel shifts, so 49 combinations are needed to calculate the final loss.

## 3. Results

### 3.1. Comparisons

We compare 3DWDSRnet to the top methods at the moment the investigation was performed (February 2020): DeepSUM [21] and HighResnet [22] (Table 1). It is worth noting that the Kelvin ESA Competition is still open to teams that want to try their solution in a post-mortem leaderboard. As the code of 3DWDSRnet is open to use, modify, and share, a GitHub repository [33] was provided. There are teams that at the time of writing this technical note have built upon it and are still improving the metrics [34].

A description of the compared methods in Table 1 follows:

- DeepSUM: It performs a bicubic upsampling of images before feeding the network. It uses convolutional layers with 64 filters, a $96 \times 96$ patch size, and 9 frames. When using this approach, higher specifications are needed because of increasing memory cost, making it impossible to train on low specification equipment. It was trained on a Nvidia 1080Ti GPU.
- Highres-net: This method upscales after fusion, so memory usage is reduced. However, it still needs 64 convolutional filters, 16 frames, and $64 \times 64$ patches to reach the best score. Scores are improved by averaging the outputs of two pretrained networks.
- 3DWDSRnet: Our method follows the Highres-net approach of upscaling after fusion, but achieves similar scores using less than half of the image frames (7), half the patch size ($34 \times 34$), and 32 convolutional filters. Moreover, there is no need to average two models to obtain these results. Figure 5 shows a real prediction done by our best model.

**Table 1.** Scores obtained in the Kelvin ESA Competition public leaderboard (February 2020). Scores are normalized by baseline; less is better. Symbol (+) refers to the memory requirements based on the number of frames, the image patch size, and the filters used.

| Method | Patch | Frames | Loss | Normalization | Score | Memory Requirement |
|---|---|---|---|---|---|---|
| DeepSUM | $96 \times 96$ (bicubic) | 9 | cMSE | Instance | 0.94745 | +++ |
| HighResnet | $64 \times 64$ | 16 | cMSE | Batch | 0.94774 | ++ |
| **3DWDSRnet (ours)** | $34 \times 34$ | 5 | cMAE | Weight | 0.97933 | + |
| **3DWDSRnet (ours)** | $34 \times 34$ (aug) | 5 | cMAE | Weight | 0.96422 | + |
| **3DWDSRnet (ours)** | $34 \times 34$ (aug) | 7 | cMAE | Weight | **0.94625** | + |

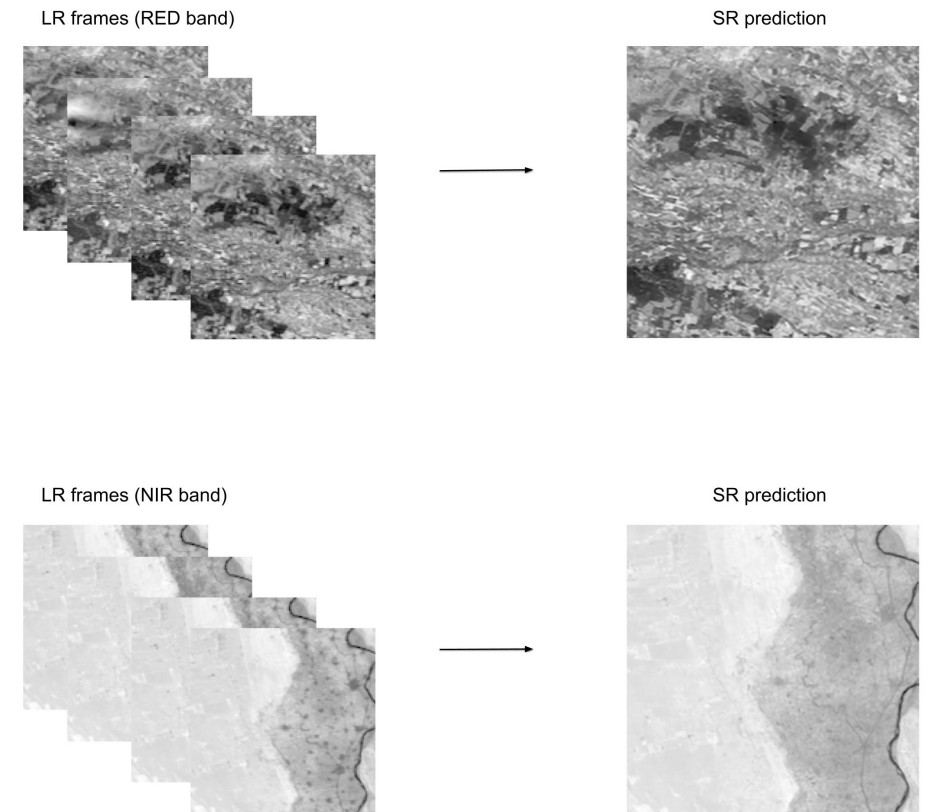

**Figure 5.** RED and NIR band SR example. The SR predicted images were obtained from our best 3DWDSRnet model (Table 1). The robustness of the method to artifacts can be seen in the example of the RED band, where the second frame (from left to right) shows clouds in the upper left that do not affect the SR prediction.

## 4. Conclusions

In this technical note, we present 3DWDSRnet, a deep learning approach to MFSR that can be used on a low-budget computer and achieves scores that are among the best in ESA's Kelvin Competition. It up-scales multiple low resolution image frames (pre-registered by phase correlation) from the PROBA-V dataset, through a two path network. One path fuses the frames using concatenated 3D WDSR blocks, and the other applies 2D WDSR Conv blocks to the pixel mean of the input frames. This method shows that by applying a simple combination of pre-processing steps (frame registration, random frame permutation, clearest frame selection) and a common CNN architecture, a remarkable improvement in the quality of the abundant low-cost satellite images available can be obtained.

## 5. Discussion

The results address some interesting insights about the common methods used in MFSR. It is shown that not always the more complex and memory consumption architectures are indeed the best ones. Sometimes, a simple model, but with the correct parameters, performs better. For example, in our method, increasing the frames from five to seven shows an increased performance (Table 1). Moreover, tweaking the data outside the neural network can improve the metrics even more. A simple method such as frame permutation for data augmentation shows a consistent growth in the score.

This makes us wonder if the money and work hours invested in designing the architecture of new neural networks as a general problem-solving approach is always a good choice. We point out,

instead, the need to develop, in addition, algorithms optimized for every need. The latter could be used by teams or individuals with less hardware resources. The access of developing countries to the latest advances in hardware is not always possible, so in order to democratize the access to AI worldwide, more research should be done on accessible, but equally efficient models.

This technical note serves as a base to continue improving the 3DWDSRnet method. Some possible directions to explore are to:

- Further investigate data augmentation methods to benefit from multiple frames such as more interesting permutations, inserting of pixel variations simulating clouds, and changes in image color, brightness, contrast, etc.
- Ensemble results from multiple models as done in Highres-net [22].
- Try different patch sizes and see how this affects the performance.

**Funding:** This research received no external funding.

**Acknowledgments:** Joaquín Padilla Montani, Fernando G. Wirtz, and Ricardo A. Dorr for general corrections.

**Conflicts of Interest:** The author declares no conflict of interest.

## Abbreviations

The following abbreviations are used in this manuscript:

| | |
|---|---|
| SR | Super Resolution |
| MISR | Multi-Image Super Resolution |
| MFSR | Multi-Frame Super Resolution |
| CNN | Convolutional Neural Network |
| Conv | Convolutional |
| AI | Artificial Intelligence |
| MAE | Mean Absolute Error |
| MSE | Mean Squared Error |
| cPSNR | clean Peak Signal-to-Noise Ratio |

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
