# Peer review of "Satellite Image Multi-Frame Super Resolution Using 3D Wide-Activation Neural Networks"

_remotesensing, doi:10.3390/rs12223812_

Round 1

Reviewer 1 Report

Very nice article and well-timed with today's interest in small satellites and with machine learning.  I liked the brevity of the article.

Author Response

Thank you very much for taking the time to review the article and for your positive feedback.

Reviewer 2 Report

The paper highlights the transfer and finetuning of a modern video superresolution neural network architecture (3DSRnet) towards the application domain of multiframe super-resolution by utilization of WDSR blocks. The approachs shows its merit on the PROBA-V benchmark set of the European Space Agency, where it shows to be competetive. The technical aspects of this application are outlined and an emphasize is giving on the simplicity and low computational cost in comparison to other approaches to the same data.

This paper is self-contained and defines clearly its goal. Necessary references to literature is highlighted and the corresponding problem at hand (the data-fusion of multiple low resolution images to a high resolution target) is explained correctly. No unnecessary information is provided and it is a very clear and easy read. Emphasis is given to explaining how an established super-resolution network can be adapted. Design choices are motivated by an experimental approach and observations of the author. Supplementary, the author provides an online open software repository which contains the full pipeline from dataloading, preprocessing, training to evaluation. This reviewer took the liberty to actually check whether this code holds up to the promise of running on low-budget hardware by testing it on his own poor machine. Apart from some minor modifications, the pipeline was in order and produced expected results. Thus, I have no reason to doubt that the results in this paper are solid and could be reproduced by peers with a little dedication.

While the proposed approach of the author is no longer on top of the leaderboard of the (post)competition, it holds its own against other approaches in quality while being arguably more memory-efficient. The weak point of this work is a lack in novelty, as pre-established neural network blocks are re-essembled in a straightforward fashion. The ablation study in this work is minimal, highlighting only 3 versions of the proposed architecture in the table while giving a few more details on contributing factors in the text. This area could surely be expanded, as the author himself admits some further open research directions, especially in the direction of data augmentation. Although the approach lacks in originality, its simplicity is appealing and its effectiveness has been clearly demonstrated.

Here are some concrete suggestions to the author:

Introduction: The overall motivation of the work is good and relevant references are given. However, the aspect of transfer-rates could be further elaborated (maybe 1 or 2 references), as this is becoming one of pending challenges for EO satellites.

Results: Indicating the memory requirements with "+++", "++" and "+" can mean anything. It would be better to give a quantitative indication on the memory required rather than a qualitative (although the reader might "guess" about this given the patchsizes and frame-counts).

It would not harm to include some examples of the prediction (SR images) of the trained model. Figure 4 does not suffice to show any differences between the images.

The provided code is functional, but certain library dependencies (click, pandas, scikit-learn) should be mentioned in the repository.

Traditionally, a paper should end with conclusions, while the author ends in a discussion section with an outlook. It would be better to separate the lessons learned from the more speculative meaning discussion and future suggestions.

Author Response

Thank you very much for taking the time to review the article. Please see the attachment for responses.

Reviewer 3 Report

General Comment

The submitted technical report proposes a simple 3D convolutional neural network (CNN) architecture with Wide Activation Super-Resolution (WDSR) blocks and a frame-based permutation technique to enhance the spatial resolution of satellite images. The proposed DCNN architecture along with the permutation technique try to resolve the complex nonlinear mapping between LR image and corresponding HR image.

In general, the submitted manuscript is well organized. Comments about several issues in the current manuscript are given as below.

Comment 1

In section 2, the author mentions that for each LR and HR image there is a mask that provides more reliable pixels for super-resolving. However, the author does not provide any information about the properties of the mask or any clarification about the pre-processing steps to discriminate reliable pixels.

Comment 2

The author believes that there are similarities between applying SR techniques on video frames and on remote sensing data of high temporal resolution to enhance the quality of a single LR video frame or a single LR satellite image, where information from neighboring LR frames are used to enhance the quality of the predicted super-resolved image. My biggest concern is regarding the different interpretations for HR images among computer vision and remote sensing experts. In RS, one-pixel displacement in LR satellite image causes errors of large magnitude in identifying the location of an object. When multi-frame LR images, with at least one-pixel shift and at most several pixels shift (depending on the number of neighboring frames) is used to predict pixels in the HR image, such pixel displacement can be more concerning. The author should clarify that how this MFSR approach is robust against such pixel displacement in predicted SR images.

Comment 3

It is suggested that the author provides some citations in section 1 regarding MFSR techniques applied on remote sensing data to support the MFSR methodology to enhance spatial resolution of RS images.

Comment 4

Caption in Figure 2, “…. To obtain the final RH frame.”, RH should be replaced by HR.

Comment 5

In subsection “Preprocessing and Data Augmentation”, the dirty pixels and their properties need to be defined. Furthermore, the author should briefly explain the process to select the first seven cleanest frames.

Comment 6

In line 143, it would be helpful to potential readers how the parameter, d, “maximum expected shift between images” is determined and if the loss function is able to handle both the magnitude and the direction of the shift with respect to the HR image.

Author Response

Thank you very much for taking the time to review the article and your feedback. Please see the attachment for responses.

Round 2

Reviewer 3 Report

The authors performed the corrections.